# All rivers run into the sea:
# Unified Modality Brain-like Emotional Central Mechanism

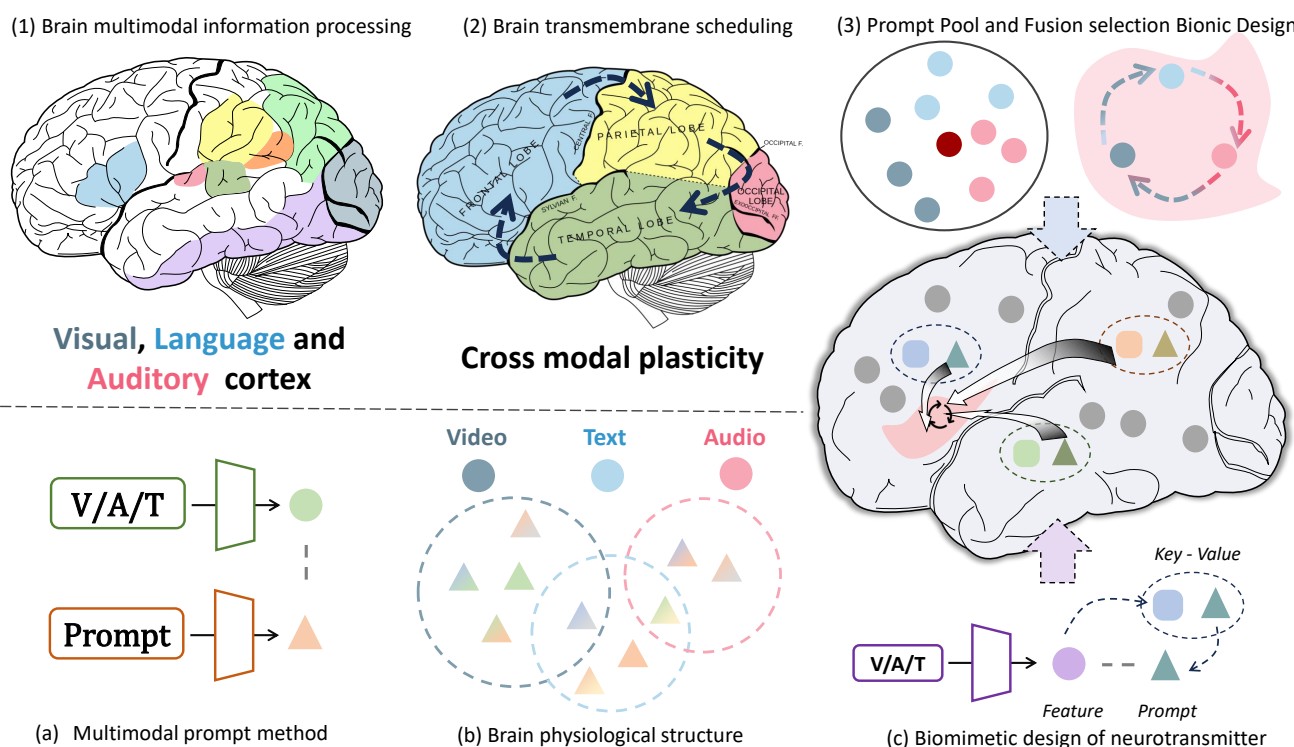

(1) Brain multimodal information processing  (2) Brain transmembrane scheduling  (3) Prompt Pool and Fusion selection Bionic Design

**Visual, Language and Auditory cortex**

**Cross modal plasticity**

(a) Multimodal prompt method  (b) Brain physiological structure  (c) Biomimetic design of neurotransmitter

Figure 1: The research motivation of our designed UMBEnet. (1) and (2) illustrate the theory of cross-modal plasticity, wherein the human brain can recruit neurons from the center of the absent modality to enhance the analytical capabilities of the remaining modalities when a certain modality is missing. Inspired by this, we have designed a Dual-Stream (DS) structure as shown in (3), where the Prompt Pool contains prompts of different modalities, which after extraction are fused with inherent prompts. This mechanism simulates how neurons of different modalities are activated and eventually integrated for analysis in the emotional center. (a) shows how in traditional multimodal methods, prompts correspond one-to-one with features without any directional flow of information. (b) illustrates features first activate neurons corresponding to their modality, and neurons can be activated by multiple modalities simultaneously, after which the activated neurons process the feature information in an integrated manner in the human brain. Inspired by this, we introduce a biomimetic design in (c) that imitates neurotransmitter activation, where multimodal features first activate the corresponding modality's prompts through key-value pairs. These prompts are then processed collectively before being used for comparative prediction. (c) in conjunction with the Prompt Pool in (3), culminates in a structure analogous to that shown in (b). Notably, the actual utilization involves the 'value' rather than the 'key', mirroring the process where neurons, once activated by neurotransmitters, propagate electrical signals.

## ABSTRACT

In the field of affective computing, fully leveraging information from a variety of sensory modalities is essential for the comprehensive understanding and processing of human emotions. Inspired by the process through which the human brain handles emotions and the theory of cross-modal plasticity, we propose UMBEnet, a brain-like unified modal affective processing network. The primary design of UMBEnet includes a Dual-Stream (DS) structure that fuses inherent prompts with a Prompt Pool and a Sparse Feature Fusion

(SFF) module. The design of the Prompt Pool is aimed at integrating information from different modalities, while inherent prompts are intended to enhance the system's predictive guidance capabilities and effectively manage knowledge related to emotion classification. Moreover, considering the sparsity of effective information across different modalities, the SSF module aims to make full use of all available sensory data through the sparse integration of modality fusion prompts and inherent prompts, maintaining high adaptability and sensitivity to complex emotional states. Extensive experiments on the largest benchmark datasets in the Dynamic Facial Expression Recognition (DFER) field, including DFEW, FERV39k, and MAFW, have proven that UMBEnet consistently outperforms the current state-of-the-art methods. Notably, in scenarios of Modality Missingness and multimodal contexts, UMBEnet significantly surpasses the leading current methods, demonstrating outstanding performance and adaptability in tasks that involve complex emotional understanding with rich multimodal information.

## CCS CONCEPTS

• **Computing methodologies** → **Computer vision**.

## KEYWORDS

Affective Computing, Modality Missingness, Cross-Modal Plasticity, Dynamic Facial Expression Recognition

## 1 INTRODUCTION

The fusion of multimodal sensory information forms the cornerstone of human perception and cognition [12]. However, the realms of multimodal fusion and Modality Missingness pose significant challenges within the field of affective computing. Modality Missingness refers to the unavailability of multimodal information in real-world affective computing tasks, where only certain specific modalities can be utilized. In affective computing, Modality Missingness poses significant challenges to methods based on text[18, 32, 46], audio[6, 33], video[34], and the like. Even for multimodal approaches[1, 26], performance can significantly deteriorate in environments where modalities are absent. The unpredictable nature of Modality Missingness can drastically diminish the accuracy and robustness of computational models designed to interpret human emotions. This issue becomes even more pronounced in dynamic and unstructured environments, where the availability of multimodal data may be inconsistent.

Recent advancements in neuroscience have underscored the phenomenon of cross-modal plasticity [4, 7, 9], wherein the lack of input from one sensory modality leads to compensatory enhancements in the processing capabilities of others[10, 13, 15],. For instance, in the case of individuals with vision loss, certain neurons in the primary visual cortex, which would typically process visual stimuli, can be recruited by other sensory modalities to enhance their processing capabilities [16, 35, 38]. This observation aligns with the everyday experience of heightened auditory and tactile sensitivities in individuals who are blind. Such insights reflect the brain's remarkable ability to reorganize and optimize sensory processing under constrained conditions [39]. In the field of emotional understanding, Albert Mehrabian proposed the 7%-38%-55% rule[31], which suggests that 7% of emotional information is conveyed through verbal expression, 38% through tone of voice,

and 55% through facial expressions, highlighting the predominant role of visual information.

Dynamic Facial Expression Recognition (DFER) represents a pivotal downstream task within the realm of affective computing, placing a greater emphasis on the understanding of visual emotions. This is consistent with Albert's theory. Additionally, a distinct advantage within the DFER domain is that the datasets inherently provide raw visual information, as DFER necessitates original facial data for the recognition of emotions. In contrast, other visual-based affective computing tasks often process their data into features, significantly constraining us to test our methods in scenarios that more closely resemble real-world conditions. Based on this, we explore the avenues of Modality Missingness and multimodal fusion within the context of multimodal DFER tasks.

Inspired by these insights, as well as the use of multiple prompts in continuous learning tasks, we propose a **U**nified **M**odality **B**rain-like **E**motional net for affective computing, named **UMBEnet**, marking a paradigm shift in the challenge of unified modal emotional understanding. UMBEnet's design draws inspiration from the brain's ability to reconfigure and augment its processing capabilities in response to sensory deprivation, integrating a Dual-Stream (DS) structure and a Sparse Feature Fusion (SFF) module. We have designed a Prompt Pool (First Stream) that employs trainable multiple prompts and a mechanism that simulates neural impulse transmission, capable of fully harnessing multimodal information, and blending it with inherent prompts (Second Stream) that store emotional information to form a Dual-Stream mechanism. Prompts are considered analogous to neurons in the brain, capable of storing a certain amount of information. The design of the Prompt Pool aims to emulate the brain's ability to select and interpret information in varying contexts, particularly in environments where modalities are absent. The design of inherent prompts seeks to emulate the activation of the amygdala, the emotional center of the brain, integrating multimodal information for judgment. SFF introduces a mechanism akin to actual neural impulse transmission. Considering that most features carry low-value information for transmission, sparse matrix fusion mimics the sparsity of neural impulse transmission in the brain[2], sparsely blending the prompts from the dual-stream mechanism. Inspired by the 7%-38%-55% rule, we designed an imbalanced multimodal encoder within UMBEnet. This encoder allocates a larger proportion of parameters to visual processing, reflecting the leading role of visual information in emotion recognition, while reducing the parameters for textual and auditory processing.

UMBEnet introduces several innovations to affective computing, with three main contributions:

- We innovatively combined inherent prompts with a Prompt Pool to design and introduce a **Dual-Stream (DS)** structure. This dual-stream approach ensures a comprehensive understanding of human emotions by leveraging the strengths of different sensory modalities.
- Our framework includes a **Sparse Feature Fusion (SFF)** module, optimizing the use of available sensory data. By sparsely integrating modality fusion prompts with inherent prompts, this module allows for the efficient fusion of

multimodal information, significantly enhancing the robustness and accuracy of emotion recognition across diverse scenarios.

- Drawing inspiration from the neural anatomical structure of the human brain, UMBEnet employs a **Brain-like Emotional Processing Framework (BEPF)**. This biomimetic design closely mimics the human brain's emotional center, not only offering a more natural and effective way of understanding emotions but also improving system performance in emotion recognition tasks.

These contributions collectively enable UMBEnet to consistently outperform existing SOTA methods in DFER domain, particularly in scenarios involving multimodality and modal absence.

## 2 RELATED WORK

### 2.1 Dynamic Facial Expression Recognition

In Dynamic Facial Expression Recognition (DFER), the trend has shifted from static to dynamic analysis, emphasizing temporal dynamics in expressions [44]. This shift is propelled by deep learning advancements, with 3D Convolutional Neural Networks (C3D) capturing both spatial and temporal data dimensions [42], and Transformers like Visual Transformers (ViT) excelling in feature extraction and sequence processing [11]. These innovations offer refined emotion recognition, addressing the complexities of video data and long-term dependencies in facial expressions. Furthermore, contrastive learning methods, exemplified by CLIP[37] and its enhancement, CLIPER [24], have forged synergies between visual and textual modalities, elevating recognition precision across diverse scenarios.

Our approach diverges from the aforementioned methodologies by adopting a novel brain-inspired architecture, deviating from the conventional design philosophies of DFER methods. Our approach, particularly the prompt pool design and activation mechanism, innovatively addresses the challenge of missing modalities, merging inherent prompts within our SSF framework. This fusion not only aids in managing multimodal data but also enriches the interpretability of DFER systems.

### 2.2 Modality Missingness

Affective computing strives to empower computers with the ability to recognize and understand human emotional states by integrating information from various sources such as voice, facial expressions, and physiological signals. The absence of one or more of these modalities, a situation known as modality missingness, complicates the task significantly [3]. To tackle such challenges, recent studies have delved into multimodal DFER methods, focusing on innovative strategies like modality compensation and data fusion to handle the absence or corruption of critical sensory data [47]. However, at present, the methods of modality missingness in the field of emotional computing are very limited, and most of them can not achieve good performance.

Our approach introduces the challenges of Modality Missingness and multimodal fusion into the realm of DFER, aiming to fully leverage all available modal information in a manner that aligns with the decision-making structure of the human brain's emotional center.

### 2.3 Cross-Modal Plasticity and Brain Sciences

The brain's capability to process multimodal information offers critical theoretical insights for the field of affective computing. Anatomical discoveries indicate that the emotional center of the brain is located in the amygdala [22]. Humans gather multimodal information through photoreceptors in the eyes, hair cells in the ears, etc., and process this information through primary visual and auditory centers, which are then integrated by the amygdala and analyzed by the cerebral cortex [40]. The theory of cross-modal plasticity[4] also proves that the primary center can recruit neurons from other centers for analysis [8, 28, 29]. These anatomical insights are crucial for designing algorithms capable of mimicking the brain's ability to process complex emotional information [17], especially in situations where data are missing or distorted.

## 3 METHOD

UMBEnet employs a brain-like emotional processing framework, consisting of two key components: DS and SFF. In the sections below, we detail the key design and functionality of each component.

### 3.1 UMBEnet's Brain-like Structure

UMBEnet initially consists of a series of transformer-based encoders acting as feature extractors, processing different modal inputs separately. Let the original inputs for visual, textual, and audio modalities be $X_v, X_t, X_a$ respectively. The corresponding modal encoders $E_v, E_t, E_a$ transform these inputs into high-dimensional feature vectors, such that $F_v, F_t, F_a \in \mathbb{R}^{B \times F}$. :

$$F_{modal} = E_{modal}(X_{modal}), \tag{1}$$

where $E_{modal}$ represent the encoder functions for visual, textual, and audio inputs, respectively.

Next, these feature vectors are fed into an adaptive self-attention module for information fusion. In the adaptive self-attention module, features from each modality are weighted and combined to generate a comprehensive multimodal embedding $F_{fusion}$:

$$F_{fusion} = \text{AdaptiveFusion}(F_v, F_t, F_a). \tag{2}$$

In the adaptive self-attention mechanism AdaptiveFusion, we utilize the Query (Q), Key (K), and Value (V) mechanisms of the Transformer for information processing and the module requires an additional modal mask matrix:

$$\text{Attention}(Q, K, V) = \text{softmax}\left(\frac{Q_i K^T + M}{\sqrt{d_k}}\right) V, \tag{3}$$

where $Q$, $K$, and $V$ represent the Query, Key, and Value matrices, respectively, and $d_k$ is the dimensionality factor for appropriate scaling. $M$ represents the modal mask matrix that is added to the scaled dot products of the queries and keys. Through self-attention mechanism and the modal mask matrix, our model can generate specialized attention weights for each modal feature, thereby optimizing the fusion process and ensuring that meaningful final outputs $F_{fusion}$ can still be produced even when some modal information is missing.

After fusion by the adaptive self-attention module, the synthesized embedding $F_{fusion}$ is sent into a predefined Prompt Pool with the purpose of finding a set of prompts that best match the current

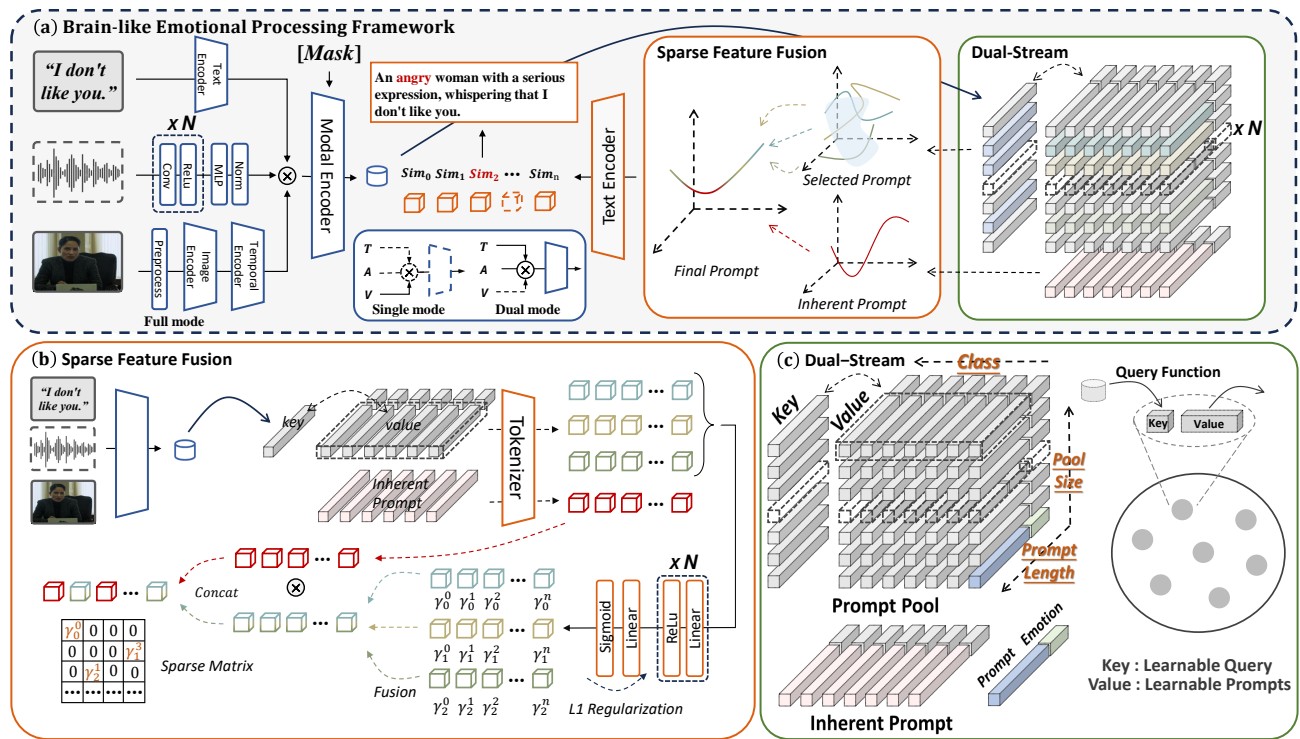

Figure 2: Overall architecture of UMBEnet. Figure 2a shows a brain-like emotional processing framework (BEPF). The left half of the diagram represents an unbalanced encoder, with the majority of parameters dedicated to visual encoding, while the right half shows the activated prompts. After multimodal information is encoded by the unbalanced encoder, it activates multimodal prompts in the Prompt Pool via a mapping function. These prompts, together with inherent prompts, undergo sparse feature fusion, and their similarity with the multimodal information is calculated. Figure 2b illustrates the structure of the Sparse Feature Fusion (SFF), including how multimodal prompts are merged with inherent prompts. Figure 2c presents the architecture of the Dual-Stream (DS), with the left side showing the actual structure and the right side providing a flattened perspective to concretely understand the Prompt Pool and its activation mechanism.

situation. Let the Prompt Pool be $P = \{p_1, p_2, ..., p_n\}$, the selection of the most matching set of prompts can be represented as:

$$P^* = _{\{p_i\} \subset P} \cos(F_{fusion}, E_p(e_i)), \quad (4)$$

Where $E_p$ represents the Key-Value pair, $e_i$ represents the learnable embedding, and cos denotes the cosine similarity function. In the Prompt Pool, the process involves searching for the embedding most similar to the current multimodal input information, and then using that embedding as the Key to find the corresponding prompt as the Value. This process ensures that the selected prompt can be decoupled from the current multimodal input information and can also learn information missing from the modalities.

Finally, the prompts selected from the Prompt Pool are concatenated with original learnable prompts to form the final prompt representation $P_{final}$. Thereafter, $P_{final}$ together with $F_{fusion}$ are used to compute the contrastive loss to identify the closest emotion category:

$$\mathcal{L}_{contrast} = -\log \frac{\exp(\cos(F_{fusion}, E_p(P_{final})))}{\sum_j \exp(\cos(F_{fusion}, E_p(p_j)))}, \quad (5)$$

Where $j$ traverses all possible category prompts. By minimizing the contrastive loss $\mathcal{L}_{contrast}$, UMBEnet learns to accurately map multimodal inputs to their corresponding emotional categories.

## 3.2 Dual-Stream structure (DS)

The Prompt Pool $P$ contains a series of predefined textual prompts, each aimed at representing a specific emotional state or scenario. We define the Prompt Pool as $P = \{p_1, p_2, ..., p_n\}$, where $n$ is the size of the Prompt Pool, and each $p_j \in P$ represents a specific emotional scenario or state. Each $p_j$ can be further represented as a set of Key-Value pairs, i.e., $(k_j, v_j)$, where $k_j$ and $v_j$ respectively represent the key and value. In our framework, the value $v_j$ corresponds to a piece of prompt, while the key $k_j$ is used to associate the prompt with a specific state.

In Figure 3, within the Prompt Pool, unimodal features correspond to unimodal prompts, while multimodal fused features correspond to multimodal prompts. This mechanism enables our model to fully capitalize on multimodal information, effectively addressing the challenges posed by missing modalities.

The value part $v_j$ of each prompt $p_j$ is a sequence of tokens of length $L_p$, embedded into the same embedding space $D$ as the multimodal features $F_{fusion}$, expressed as:

$$v_j = [v_{j1}; v_{j2}; ...; v_{jL_p}], \tag{6}$$

$$v_{ji} \in \mathbb{R}^D, \quad \text{for } i = 1, 2, ..., L_p. \tag{7}$$

Here, $v_{ji}$ represents the embedding vector of the $i$-th token in $v_j$.

In UMBEnet, the function of the Prompt Pool is not merely to provide a fixed set of textual collections for the model to choose from. By adopting a learnable key-value pair structure, we allow the model to dynamically select the most matching prompt while dealing with a specific multimodal input. This matching process can be realized by calculating the similarity between the input features $F_{fusion}$ and each key $k_j$, then selecting the prompt with the highest similarity for subsequent processing.

$$j^* =_j \cos(F_{fusion}, k_j), \tag{8}$$

$$p^* = v_j, \tag{9}$$

Where cos denotes the cosine similarity function, $j$ is the index of the prompt most matching with the fused feature $F_{fusion}$, and $p^*$ is the value of the selected prompt.

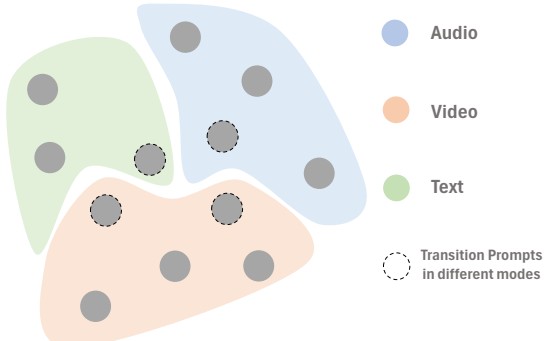

**Figure 3: Demonstration of the Prompt Pool's functionality in processing unimodal and multimodal information.**

Figure 4 illustrates how key-value pairs simulate the mechanism of neural impulse transmission: Neurotransmitters are released by one neuron, bind to receptors, and transmit neural impulses, where the content of transmission is not the neurotransmitter itself but transforms into electrical signals. Inspired by this, both our key and value are set to be trainable. The key is trained to align with the query's latent space, while the value is trained to learn modal information. Similarity between query and key is computed through different mapping functions, and the corresponding value is outputted.

Through this mechanism, the key-value pairs in the Prompt Pool make the connection coupled, not only enhancing the interpretability of the model but also improving its adaptability to different states. This design allows UMBEnet to respond more flexibly and accurately when facing complex multimodal emotion recognition tasks.

The design of the Prompt Pool ($P$) and the concatenation with inherent prompts aim to mimic the brain's strategy of activating different neurons for varying tasks, particularly under Modality Missingness. For a set of prompts, the process of selecting and

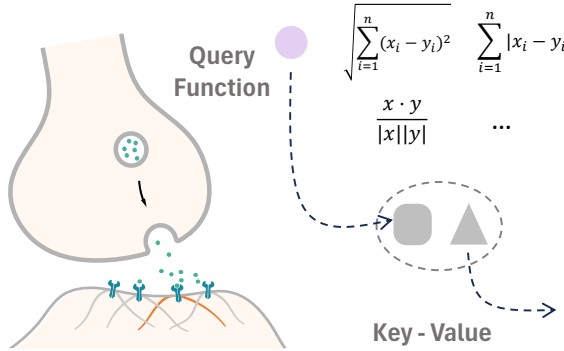

**Figure 4: The operation of the activation mechanism modeled after neural impulse transmission is depicted on the left. In this process, neurotransmitters, once received by receptors during transmission, convert not into the neurotransmitters themselves but into electrical signals, analogous to a specialized key-value pair system where receptors and electrical signals correlate. The top right corner illustrates the query function, representing selectable mapping functions within this framework. This neural-inspired approach provides a biomimetic method for prompt activation, reflecting the intricacy and efficiency of neural communication in UMBEnet's architecture.**

concatenating prompts can be formalized as:

$$P_{selected} =_{p_i \in P} \cos(F_{fusion}, E_p(p_i)),$$

where $E_p(p_i)$ represents the embedding of the $i$-th prompt, and cos denotes the cosine similarity between the fused features $F_{fusion}$ and the embedded prompt. This selection process is crucial for adaptively responding to different emotional contexts.

### 3.3 Sparse Feature Fusion (SSF)

During the modal fusion stage, we employ the SSF mechanism to further process and synthesize the prompts selected from the Prompt Pool. Suppose the set of $tok_p$ most matching prompts selected from the Prompt Pool forms $P^* = \{p_1^*, p_2^*, \ldots, p_{tok_p}^*\}$, where each $p_i^*$ is a prompt value corresponding to a specific emotional state selected through the aforementioned process.

Next, these selected prompts are sent into a sparse feature fusion process to capture their interactions and their relations with the original multimodal information:

$$P' = \text{SparseFeatureFusion}(P^*), \tag{10}$$

Where $P'$ represents the set of prompts processed by the SSF mechanism.

The SSF is implemented through a series of linear layers and ReLU activations, finalized with L1 regularization for sparsity, followed by another linear layer and a sigmoid function to compute a sparse matrix. Specifically, for a given input $X$, the sparse feature fusion can be expressed as:

$$X_{sparse} = \sigma(W_2 \cdot \text{ReLU}(W_1 \cdot X + b_1) + b_2), \tag{11}$$

$$\sigma(x) = \frac{1}{1 + e^{-x}}, \tag{12}$$

and $W_1$, $W_2$ are weight matrices, $b_1$, $b_2$ are bias terms, and $\sigma$ is the sigmoid function ensuring the output is in the $(0, 1)$ range, simulating the sparsity in neural activations.

During the sparse feature fusion process, L1 regularization is applied to the weight matrix $W_1$ to promote sparsity. The new cost function incorporating L1 regularization for the weight matrix $W_1$ is given by:

$$L(W_1) = L_{ori} + \lambda \sum_i |W_{1,i}| \tag{13}$$

where $L(W_1)$ is the cost function after regularization, $L(ori)$ is original loss, $\lambda$ is the regularization parameter, and $W_{1,i}$ represents the elements of the weight matrix $W_1$. The regularization term encourages the sparsity in $W_1$ by penalizing the absolute values of the weights.

Then, the obtained set of prompts $P'$ is concatenated with a set of inherent learnable prompts $P_{inherent}$, forming a comprehensive set of prompts $P_{combined} = P' \oplus P_{inherent}$, where $\oplus$ denotes the concatenation operation. This process aims to combine dynamically selected prompts with fixed, task-related inherent prompts to enhance the model's expressiveness and adaptability.

Finally, we use the contrastive loss to optimize the model, ensuring the selected set of prompts matches correctly with the multimodal input information. The model outputs the final emotional classification results by evaluating the match between $F_{fusion}$ and each category-corresponding comprehensive set of prompts $P_{combined}$:

$$y_{pred} =_i \text{sim}(F_{fusion}, P^i_{combined}). \tag{14}$$

In this way, UMBEnet can use multimodal information and rich semantic prompts for accurate emotion recognition.

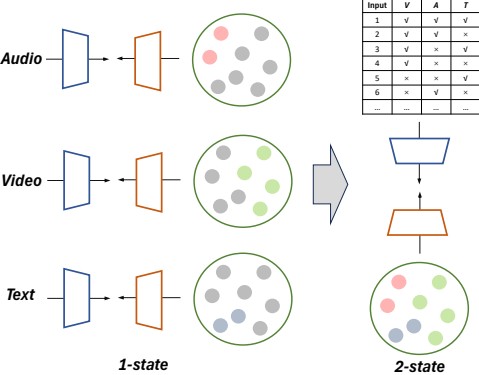

**Figure 5: The training strategy of UMBEnet unfolds in 2-stages: First, prompts are trained with unimodal inputs; Second, prompts activated in the first stage are aggregated and retrained to enhance integration and responsiveness.**

## 3.4 Training Strategy

As illustrated in Figure 5, UMBEnet's comprehensive training strategy begins by independently training prompts within the Prompt Pool using unimodal information. Initially, the system inputs audio (A), visual (V), and textual (T) modalities separately to tailor prompts specific to each modality. This first phase of training ensures that the prompts are finely tuned to respond to the unique features of each input type.

In the subsequent phase, prompts that were activated in the first stage are transferred to a new Prompt Pool. Here, the system undergoes training with randomly missing modal information, simulating scenarios where certain modalities may be miss or incomplete. This two-tiered training approach allows the model to delve deeply into the nuances of modal information, effectively leveraging the partial knowledge obtained from each modality to compensate for any missing data. Through this strategy, UMBEnet enhances its capacity to interpret complex emotional prompts by becoming proficient at drawing inferences from incomplete or asymmetrically available data. The employment of this training regime not only promotes the robustness of the system in handling real-world scenarios where multimodal data might not always be complete but also aligns with the cognitive flexibility inherent in human emotional understanding, where inferences are often drawn from partial information.

## 4 EXPERIMENTS

### 4.1 Experimental Setup

All experiments in this study were conducted in a hardware environment with the following specifications: two NVIDIA GeForce RTX 3090 graphics cards and a computer equipped with an Intel(R) Xeon(R) CPU 5218R @ 2.10GHz. During the model training process, we chose the Adam optimizer as our optimization algorithm. The initial learning rate was set to 0.002, and we adopted a mini-batch training approach with a batch size of 16. Furthermore, to prevent overfitting and improve the model's generalization ability, we designed a dynamic learning rate adjustment mechanism based on the performance on the validation set. Specifically, if there is no significant decrease in loss on the validation set for three consecutive training epochs, the learning rate will be reduced to 0.6 times its original value. When the learning rate drops below $1 \times 10^{-7}$, we consider the model to have reached early convergence, at which point the training process will be terminated. In addition, to further control the phenomenon of overfitting, we will take corresponding measures to adjust when the accuracy of the model on the training set is more than 80%, we will terminate the training in advance. In the experimental results, **bold** represents the best, and underline represents the second best. The confusion matrices and feature visualizations in Figures 6 and 7 showcase the exceptional performance of UMBEnet on FERV39k, DFEW, and MAFW. For more confusion matrices, please see **supplementary materials**.

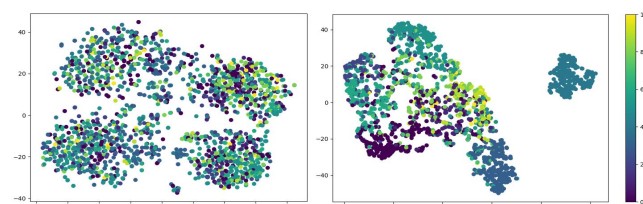

**Figure 6: Features before and after processing by the model. The left side displays features just entered into the model, scattered overall; the right side shows features before output, demonstrating improved clustering. 0-10 in the legend represents 11-class classification.**

## 4.2 Evaluation Metrics

When evaluating model performance, we adopted WAR and UAR as evaluation metrics. WAR (Weighted Accuracy Recall) refers to the weighted average of the prediction accuracy of the classification model on each class. This metric considers the sample size of each class, making it more suitable for imbalanced datasets. UAR (Unweighted Average Recall) refers to the average recall of the classification model on each class, regardless of the sample size of the class. This metric treats each class equally, thus being more suitable for balanced datasets. These two metrics provide a comprehensive assessment of the model's performance across multiple classes and are widely used evaluation metrics in the DFER domain.

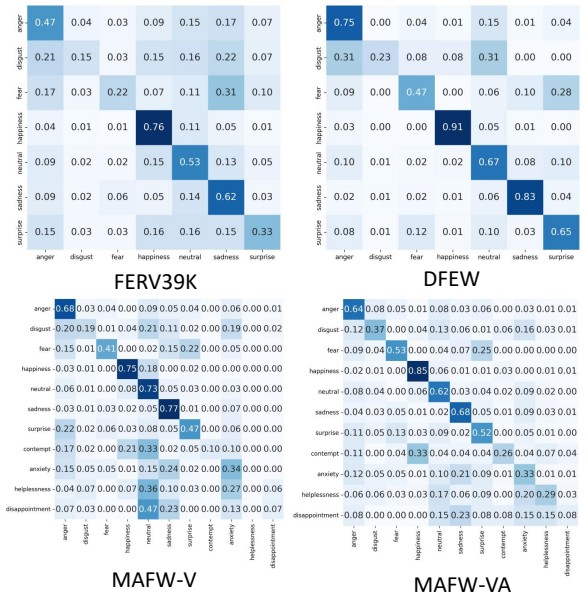

**Figure 7: Partial confusion matrices of UMBEnet on FERV39K, DFEW, and MAFW.**

## 4.3 Comparative Experiments

As shown in tables 1 and 2, to appraise the performance of our UMBEnet model in the task of emotion recognition, we carried out a suite of comparative tests on three of the largest unrestricted publicly available datasets within the DFER sphere: MAFW[27], FERV39K[44], and DFEW[21]. Notably, our approach holds its ground even against methods such as MAE that have been pre-trained on extensive data—a comparison often deemed unfair due to the considerable advantages conferred by extensive pre-training. Moreover, our method substantially outperforms the CLIP-based methods that utilize the same backbone as ours, underscoring the efficacy of UMBEnet in processing emotional data.

Among the methods that do not rely on large pre-training models, our method surpasses the SOTA AEN method by 4.56% (WAR) and 7.89% (UAR) in scenarios with missing modalities in unimodal settings, and extends the lead to 5.56% (WAR) in multimodal settings. Against SOTA CLIP-based methods with the same backbone, our method outperforms DFER-CLIP by 4.94% (WAR). Remarkably, even against the self-supervised MAE-DFER method, which has been pre-trained on extensive datasets, our method exceeds performance by 1.14% (UAR on DFEW), 0.89% (UAR on FERV39K), 0.03% (WAR on

FERV39K), and 0.73% (WAR on MAFW) across the three datasets of DFEW, FERV39K, and MAFW in unimodal scenarios. It's important to note that comparisons with self-supervised methods are often considered unfair due to their training on vast amounts of data. On multimodal datasets, our method comprehensively surpasses existing methods, with UMBEnet exceeding MAE-DFER by **6.41%** (UAR on MAFW) and **13.72%** (WAR on MAFW).

**Table 1: Overall Model Performance Comparison (UMBEnet vs. other SOTA methods on DFEW and FERV39K for 7-class classification. * represents visual and audio modal input).**

| Method | Publication | DFEW | | FERV39k | |
|---|---|---|---|---|---|
| | | UAR | WAR | UAR | WAR |
| C3D [42] | CVPR'15 | 42.74 | 53.54 | 22.68 | 31.69 |
| P3D [36] | ICCV'17 | 43.97 | 54.47 | 23.20 | 33.39 |
| I3D-RGB [5] | ICCV'17 | 43.40 | 54.27 | 30.17 | 38.78 |
| 3D ResNet18 [19] | CVPR'18 | 46.52 | 58.27 | 26.67 | 37.57 |
| R(2+1)D18 [42] | CVPR'18 | 42.79 | 53.22 | 31.55 | 41.28 |
| ResNet18+LSTM [20] | / | 51.32 | 63.85 | 30.92 | 42.95 |
| ResNet18+ViT [11] | / | 55.76 | 67.56 | 38.35 | 48.43 |
| EC-STFL [21] | MM'20 | 45.35 | 56.51 | / | / |
| Former-DFER [21] | MM'21 | 53.69 | 65.70 | 37.20 | 46.85 |
| NR-DFERNet [25] | arXiv'22 | 54.21 | 68.19 | 33.99 | 45.97 |
| DPCNet [45] | C&C'23 | 57.11 | 66.32 | / | / |
| T-ESFL [27] | AAAI'23 | / | / | / | / |
| EST | PR'23 | 53.94 | 65.85 | / | / |
| LOGO-Former [30] | ICASSP'23 | 54.21 | 66.98 | 38.22 | 48.13 |
| GCA+IAL | AAAI'23 | 55.71 | 69.24 | 35.82 | 48.54 |
| MSCM | PR'23 | 58.49 | 70.16 | / | / |
| M3DFEL [43] | CVPR'23 | 56.10 | 69.25 | 35.94 | 47.67 |
| AEN [23] | CVPRW'23 | 56.66 | 69.37 | 38.18 | 47.88 |
| *CLIP-based methods* | | | | | |
| EmoCLIP [14] | arXiv'23 | 58.04 | 62.12 | 31.41 | 36.18 |
| CLIPER [24] | arXiv'23 | 57.56 | 70.84 | 41.23 | 51.34 |
| DFER-CLIP [48] | BMVC'23 | 59.61 | 71.25 | 41.27 | 51.65 |
| *Self-supervised methods* | | | | | |
| MAE-DFER [41] | MM'23 | 63.41 | 74.43 | 43.12 | 52.07 |
| UMBEnet | / | **64.55** | 73.93 | **44.01** | 52.10 |
| UMBEnet* | / | 62.23 | **74.83** | / | / |

Tables 2 and 3 display the superior performance of our approach in the domain of multimodal DFER. DFEW and MAFW represent the two largest datasets in the field of DFER, with DFEW comprising audio and video modalities, where our method achieves SOTA results. The MAFW dataset includes three modalities: audio, video, and text. However, the text modality lacks neutral label annotations. Introducing filler noise text information leads to an artificially high accuracy in neutral classification, suggesting that the model may be learning from noise. Given the novelty of the dataset, previous DFER methods have not highlighted this issue, and the absence of confusion matrices or open-source code from those studies precludes a fair comparison. Therefore, we recalculated WAR and UAR for a 10-class scheme, excluding neutral accuracy, as shown in Table 3.

Our approach significantly outperforms the existing SOTA methods in a multimodal 11-class setting, and even with the neutral class removed in a 10-class configuration, our method still substantially surpasses the SOTA with a **7.08%** increase in UAR and a **9.33%** increase in WAR. Comparisons in the 10-class setting, with missing modalities, reveal that the most effective modality is visual, followed by textual, with audio being the least effective. It is evident that our method significantly outperforms the existing SOTA methods, both for the 11-class and the reduced 10-class configurations.

In our experiments with missing modalities in Table 3, we discovered that inferring with a full-modal model in the absence of certain modalities can achieve, or even surpass, the performance of training unimodal models from scratch. This is attributed to the two-stage training strategy employed within the full-modal model, suggesting that information from different modalities can often be cross-utilized to enhance performance. This finding underscores the efficacy of our approach in leveraging cross-modal information, thereby boosting the robustness and adaptability of the model in handling missing modal scenarios.

**Table 2: Overall Model Performance Comparison (UMBEnet vs. other SOTA methods on MAFW for 11-class classification).**

| Method | Publication | Mode | MAFW UAR | MAFW WAR |
|---|---|---|---|---|
| C3D | CVPR'15 | V | 31.17 | 42.25 |
| ResNet-18 | / | V | 25.58 | 36.65 |
| ResNet18+LSTM | / | V | 28.08 | 39.38 |
| ResNet18+ViT | / | V | 35.80 | 47.72 |
| C3D+LSTM | / | V | 29.75 | 43.76 |
| Former-DFER | MM'21 | V | 31.16 | 43.27 |
| T-ESFL | AAAI'23 | V, A, T | 33.28 | 48.18 |
| *CLIP-based methods* | | | | |
| EmoCLIP | arXiv'23 | V | 34.24 | 41.46 |
| DFER-CLIP | BMVC'23 | V | 39.89 | 52.55 |
| *Self-supervised methods* | | | | |
| MAE-DFER | MM'23 | V | 41.62 | 54.31 |
| UMBEnet | / | V | 41.00 | 55.04 |
| UMBEnet | / | V, A | **46.92** | **57.25** |

**Table 3: Performance comparison between multimode and missing mode (UMBEnet on MAFW for 11-class and 10-class classification).**

| Method | Mode | 11 class UAR | 11 class WAR | 10 class UAR | 10 class WAR |
|---|---|---|---|---|---|
| UMBEnet | V | 41.00 | 55.04 | 38.30 | 54.79 |
| | A | 13.02 | 17.34 | 15.96 | 24.93 |
| | T | 42.36 | 58.23 | 34.92 | 52.95 |
| | V+T | 49.96 | 65.95 | 45.03 | 61.28 |
| | V+A | 46.92 | 57.25 | 44.87 | 60.22 |
| | V+A+T | **53.33** | **68.03** | **48.70** | **63.64** |
| *Missing Modal Inference* | | | | | |
| UMBEnet | V | 40.29 | 53.41 | 42.86 | 59.13 |
| | A | 10.03 | 14.34 | 12.21 | 17.58 |
| | T | 44.14 | 57.55 | 38.55 | 51.67 |
| | V+T | 52.68 | 68.35 | 48.01 | 64.03 |
| | V+A | 39.57 | 53.68 | 42.12 | 59.70 |
| | V+A+T | **54.04** | **68.90** | **49.51** | **64.65** |

**Table 4: UMBEnet Hyperparameter Ablation Study (UMBEnet on MAFW for 11-class and 10-class classification).**

| Prompt Pool Set Length | Size | TopK | 11 class UAR | 11 class WAR | 10 class UAR | 10 class WAR |
|---|---|---|---|---|---|---|
| 32 | 8 | 5 | 51.95 | 63.28 | 49.56 | 61.52 |
| 32 | 16 | 5 | 52.25 | 65.69 | 47.52 | 60.97 |
| 32 | 32 | 5 | **54.22** | 66.50 | 49.65 | 61.84 |
| 64 | 8 | 3 | 53.05 | 66.01 | 48.45 | 61.41 |
| 64 | 8 | 5 | 52.69 | 65.79 | 47.99 | 61.09 |
| 64 | 16 | 3 | 48.73 | 58.81 | 48.74 | 60.22 |
| 64 | 16 | 5 | 41.31 | 52.70 | 45.45 | 60.04 |
| 64 | 32 | 3 | 47.97 | 57.45 | **50.53** | 62.33 |
| 64 | 32 | 5 | 53.33 | **68.03** | 48.70 | **63.64** |
| *Training Strategy Ablation* | | | | | | |
| 1-stage training | | | 52.85 | 66.77 | 48.13 | 62.15 |
| 2-stage training | | | **53.33** | **68.03** | **48.70** | **63.64** |

## 4.4 Ablation Study

In the ablation study, we delve into the impact of different modalities on model performance and conduct ablations on the MAFW dataset. As shown in Table 3, we ablated the effects of modalities and, thanks to the design of our unbalanced encoder, we achieved SOTA results even with missing modalities (for example, using only visual input). Moreover, as predicted, the audio modality contributes less to accuracy, which further validates the design of our unbalanced encoder. Additionally, in Table 4, we tested various hyperparameter settings on the MAFW dataset, including TOPK, pool size, and prompt length, to analyze their impact on model performance. An overview of Table 4 reveals a generally positive correlation between the size of the designed Prompt Pool and the number of chosen prompts (topk) with accuracy, and a similar positive relationship with the overall length of the prompts. Within the training strategy ablation section, we observe that the Prompt Pool, which underwent two-stage hybrid training. In our ablation experiments focused on training strategies, we discovered that our proposed two-stage transfer training approach significantly enhances performance. Even within the same task, our training strategy achieved improvements of 0.48% in UAR and 1.26% in WAR in the 11-class configuration, and 0.57% in UAR and 1.49% in WAR in the 10-class setup. It significantly outperforms the one trained in a single phase. The visualizations in Figure 8 corroborate our hypothesis.

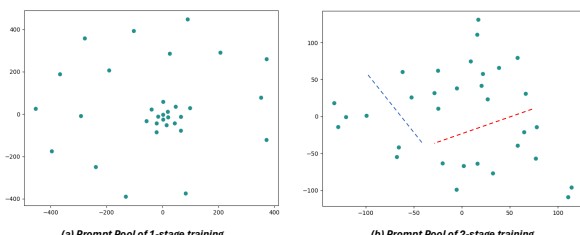

*(a) Prompt Pool of 1-stage training*     *(b) Prompt Pool of 2-stage training*

**Figure 8: Visualization of the Prompt Pool under different training strategies. Each point represents a prompt.The Prompt Pool trained with direct multimodal inputs shows trained prompts clustering together, with untrained prompts evenly dispersed around them in Figure 8a.The Prompt Pool subjected to two-stage training exhibits a partially random distribution, with a distinct modal separation marked by red and blue dashed lines in Figure 8b.**

## 5 CONCLUSION

Our work introduces UMBEnet, a novel unified modal model that departs from the paradigms of previous DFER methods, mirroring the complex neural architecture of the human brain in emotional understanding and effectively addressing the challenges of Modality Missingness and multimodal fusion. Extensive testing on leading DFER benchmarks—DFEW, FERV39k, and MAFW—has demonstrated the superior performance of UMBEnet, especially under various channel conditions or in their absence. We believe UMBEnet will be instructive to the entire multimodal community, and we will continue to explore the use of UMBEnet in other multimodal areas in future.

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
