# OpenReview forum: "All rivers run into the sea: Unified Modality Brain-Inspired Emotional Central Mechanism"
_acmmm.org/ACMMM/2024/Conference — MM2024 Oral_

### Official Review · Reviewer_zsLm · 2024-05-24

**Rating:** 4
**Confidence:** 4

**Summary:**

This paper introduces UMBEnet, a new unified modality emotion processing network that mimics the neural mechanism of the human brain in emotion understanding, specifically addressing the challenges of modality loss and multi-modality fusion. The paper describes in detail the dual-stream structure (DS) and sparse feature fusion (SSF) modules of UMBEnet and how it achieve superior performance in the field of dynamic facial expression recognition (DFER). Through extensive experiments and comparative tests, the authors demonstrate the superior performance of UMBEnet on the major benchmark datasets DFEW, FERV39K, and MAFW, especially in multi-modal and modal-missing environments.

**Strengths:**

Innovation: The paper proposes a novel brain-like architecture that combines dual-stream structure and sparse feature fusion, demonstrating innovation in the field of affective computing.

Solving practical problems: Proposing effective solutions to the problem of modal missing and multi-modal fusion, demonstrating good adaptability to real-life situations.

Ablation study: Through detailed ablation studies, the impact of different modes and hyperparameters on model performance was explored, and the effectiveness of the model design was verified.

**Limitations:**

Detailed description: The description of some technical details can be more detailed, such as specific implementation details and parameter settings in the sparse feature fusion mechanism. Clarify the specific parameters of each layer: input and output dimensions. Location and type of activation function. The specific values of the regularization parameters and the basis for their selection.
Limitations of the dataset: Although the experiments cover multiple datasets, they are mainly concentrated in the DFER field and lack validation on other affective computing tasks, which limits the generalizability of UMBEnet.

Graph Interpretation: The interpretation of certain graphs could be more explicit so that readers can more easily understand and verify experimental results. For example, Table 3 shows the experimental results in the absence of modes, but does not fully explain why the absence of certain modes has a greater impact on the results, and the performance mechanism of the model in these cases.

Open source: The paper does not mention whether open source code will be provided, which can enhance the reproducibility of results and trust in the method.

**Suitability:**

3

---

### Official Review · Reviewer_4ZEm · 2024-05-24

**Rating:** 3
**Confidence:** 3

**Summary:**

This paper proposes a Dual-Stream structure network UMBEnet for DFER, which designs a prompt-based module and a sparse feature module to make full use of all available sensory data.

**Strengths:**

1.This paper tries to introduce a biomimetic design for DFER.
2.Figure 1 is well drawn.

**Limitations:**

1. Many symbols lack definition and interpretation. What is the predefined Prompt Pool P in Line 384? Does ‘$\bigotimes$’ in Fig.2 have the same meaning?
2. Equation (3) provides a common expression for attention mechanism, but does not describe how to use it in network. Is it simple to perform self-attention for each modality? How can the fusion features be produced when some modal information is missing?
3. In psychology, Albert Mehrabian proposed the 7%38%-55% rule. However, many multimodal emotion recognition research proves that the textual modality plays the predominant role in affective computing, and the results in Tab.3 can also reflect this point. These words as the motivation may bring some confusion and misleading.
4. Due to the complexity of the proposed mothed, this paper lacks a problem definition, which increases the difficulty of understanding.
5. In Line 447, the Prompt Pool 𝑃 contains a series of predefined textual prompts. How to construct the prompts for different modalities? The presentation in Figure 3 is unclear, making it difficult to understand the author's intention.
6.The modality missingness in this work is closely related to prompt, and relevant ablation study should be designed.

**Suitability:**

3

---

### Official Review · Reviewer_sDK5 · 2024-05-25

**Rating:** 5
**Confidence:** 2

**Summary:**

The manuscript proposes a novel multimodal emotion classification method. The method basically employs dictionary learning strategy, by which the latent encodings from different modalities can be mapped to a collection of keywords (i.e., the selected prompts from the prompt pool), producing sparse representation. The detailed dictionary learning is well-established based on neuroscience, in which the dual-stream theory is realized by the sparse representation from the dictionary keywords, and the neural communication between the neurotransmitters and electrical signals are realized by the keys and values. The method outperformed 10-22 methods in three datasets and normal/modality-missing scenarios.

Overall, the method is beautiful, the illustrations and equations are eye-pleasing, and the experiment results are strong,  One flaw of the manuscript is that it lacks technical details and the proposed model is hardly reproducible (code availability was not mentioned), rendering the method either to be a ground-breaking invention or old wine in a new bottle. With this, I opt to conservatively vote for weak acceptance with somewhat confidence.

**Strengths:**

- Strong theoretical foundation. The major plus of the method is the theoretical foundation, in which several mechanisms (i.e., he dual-stream theory and neural communication) from neuroscience is employed.
- Strong comparative experiment. The comparative experiment employed 10-22 SOTAs and the proposed method outperformed them all in the three datasets and normal/modality-missing scenarios.

**Limitations:**

- Insufficient technical details and reproducibility. The writing puts too much on the theoretical elegance, resulting in technical confusions.I still have no clue as to what exactly does the model work as a whole and how exactly does the whole model look like. For example, how to obtain the long thin cubes in Fig1(c), what layers are used to produce them? What hyperparameters are used for the transformer-based encoders? I can only get inspired from the idea, and had no information to replicate the model architecture.

**Suitability:**

3

---

### Official Review · Reviewer_76LE · 2024-05-31

**Rating:** 2
**Confidence:** 4

**Summary:**

UMBEnet introduces a unique Dual-Stream (DS) structure that integrates multimodal sensory information through a Prompt Pool and a Sparse Feature Fusion (SFF) module. This approach mimics the brain's neural pathways, presenting a methodology in the realm of affective computing that deviates significantly from traditional models. The model is based on the theory of cross-modal plasticity and incorporates brain-like mechanisms that enhance the analysis capabilities by recruiting neurons across different sensory modalities. This theoretical underpinning is robust and reflects recent advancements in neuroscience, particularly concerning how the brain processes emotions.

**Strengths:**

UMBEnet introduces a unique Dual-Stream (DS) structure that integrates multimodal sensory information through a Prompt Pool and a Sparse Feature Fusion (SFF) module. This approach mimics the brain's neural pathways, presenting a methodology in the realm of affective computing that deviates significantly from traditional models. The model is based on the theory of cross-modal plasticity and incorporates brain-like mechanisms that enhance the analysis capabilities by recruiting neurons across different sensory modalities. This theoretical underpinning is robust and reflects recent advancements in neuroscience, particularly concerning how the brain processes emotions.

**Limitations:**

1)	The paper lacks detailed discussion on its performance across a broader range of affective computing tasks beyond emotion recognition from facial expressions. This could limit the understanding of the model's effectiveness in different contexts or applications that involve other types of emotional data, such as physiological signals or behavioral data.
2)	The model's complexity, particularly with the Dual-Stream structure and the Prompt Pool mechanism, might lead to high computational demands. The paper does not thoroughly address the computational efficiency or the operational requirements of UMBEnet. For real-world applications, especially in mobile or embedded systems, the resource intensity of the model could be a significant drawback.
3)	The paper should provide deeper insights into which parts of the architecture contribute most to performance improvements. This would help in understanding the necessity and efficiency of each component of the model.
4)	Missing references:
Evidential detection and tracking collaboration: New problem, benchmark and algorithm for robust anti-uav system
Anti-UAV410: A Thermal Infrared Benchmark and Customized Scheme for Tracking Drones in the Wild
Anti-uav: a large-scale benchmark for vision-based uav tracking

**Suitability:**

2

---

### Meta-Review · Area_Chair_4ReM · 2024-06-27

**Recommendation:** Accept (Oral)
**Confidence:** 4

**Metareview:**

This work proposed a unified modal affective processing network for emotion recognition tasks and allowed the integration of information from different modalities effectively - this was demonstrated by outstanding performance and adaptability in emotion recognition tasks. Three reviews (excluding an invalid review) are positive for the contribution of this work.